# 3DILG: Irregular Latent Grids for 3D Generative Modeling

**Biao Zhang**
KAUST
biao.zhang@kaust.edu.sa

**Matthias Nießner**
Technical University of Munich
niessner@tum.de

**Peter Wonka**
KAUST
pwonka@gmail.com

## Abstract

We propose a new representation for encoding 3D shapes as neural fields. The representation is designed to be compatible with the transformer architecture and to benefit both shape reconstruction and shape generation. Existing works on neural fields are grid-based representations with latents defined on a regular grid. In contrast, we define latents on irregular grids, enabling our representation to be sparse and adaptive. In the context of shape reconstruction from point clouds, our shape representation built on irregular grids improves upon grid-based methods in terms of reconstruction accuracy. For shape generation, our representation promotes high-quality shape generation using auto-regressive probabilistic models. We show different applications that improve over the current state of the art. First, we show results for probabilistic shape reconstruction from a single higher resolution image. Second, we train a probabilistic model conditioned on very low resolution images. Third, we apply our model to category-conditioned generation. All probabilistic experiments confirm that we are able to generate detailed and high quality shapes to yield the new state of the art in generative 3D shape modeling.

## 1 Introduction

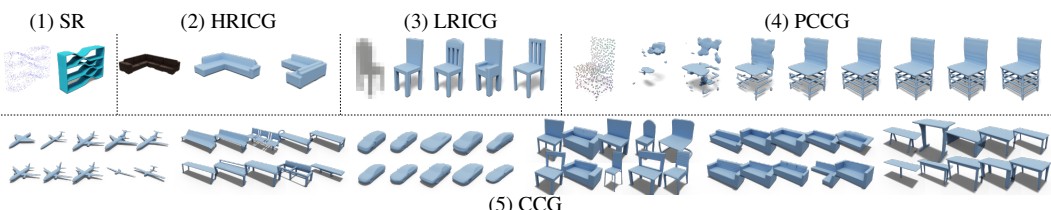

Figure 1: Irregular Latent Grids enable many applications: (1) shape reconstruction, (2) high-resolution-image-conditioned generation, (3) low-resolution-image-conditioned generation, (4) point-cloud-conditioned generation, and (5) category-conditioned generation. The data structure is especially suited for auto-regressive modeling (applications 2-5). For each of these applications the probabilistic approach enables sampling of many plausible models for a single query.

Neural fields for 3D shapes are popular in machine learning because they are generally easier to process with neural networks than other alternative representations, *e.g.*, triangle meshes or spline surfaces. Earlier works represent a shape with a single *global* latent [29, 13, 7, 36, 66]. This already gives promising results in shape autoencoding and shape reconstruction. However, shape details are often missing and hard to recover from a global latent. Later, *local* latent grids for shapes were proposed [39, 23, 3, 48]. A local latent mainly influences the shape (surface) in a local 3D neighborhood, thus perceiving shape details. However, in contrast to global latents, local latents

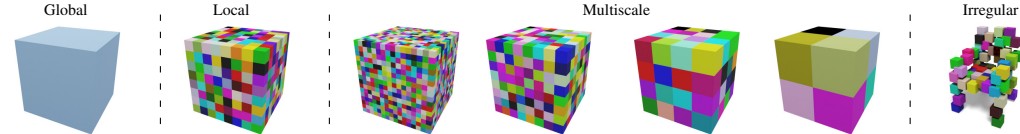

Figure 2: **Latent grids.** From left to right, we show single global latent (*e.g.*, OccNet [29]), latent grid (*e.g.* ConvONet [39]), multiscale latent grids (*e.g.* IF-Net [9]), and our irregular latent grid.

require positional information about their location, *e.g.*, as implicitly defined by a regular grid. Furthermore, *multiscale* latent pyramid grids [9, 6] provide some performance improvement over basic grids. We illustrate the three current ways of modeling neural fields in Fig. 2.

In this paper, we set out to study irregular grids as 3D shape representation for neural fields. This extends previous grid-based representations, but allows each latent to have an arbitrary position in space, rather than a position that is pre-determined on a regular grid. We do not want to give up the advantages of a fixed length representation, and therefore propose to encode shapes as a fixed length sequence of tuples $(\mathbf{x}_i, \mathbf{z}_i)_{i \in \mathcal{M}}$, where $\mathbf{x}_i$ are the 3D positions and $\mathbf{z}_i$ are the latents. The overall advantages of this representation are that it is fixed length, works well with transformer-based architectures using established building blocks, and that it can adapt to the 3D shape it encodes by placing latents at the most important positions. The representation also scales better than grid-based (and especially pyramid-based) representations to a larger number of latents. The number of latents in a representation is a factor that greatly influences the training time of transformer architectures, as the computation time is quadratic in the number of latents.

While our proposed representation brings some improvement for 3D shape reconstruction from point clouds, the improvement is very significant for 3D generative models that have been less explored. We believe that the most promising avenue for 3D shape generation is to follow recent image generative methods based on vector quantization and auto-regressive models using transformer, *e.g.* VQGAN [16]. These models operate on discrete latent vectors represented by indices [53]. During training, an autoregressive probabilistic model is trained to predict discrete indices. When doing inference (sampling), discrete indices are sampled from the autoregressive model and are decoded to images with a learned decoder. These models work well in the image domain, because a single image can comfortably be represented by medium size, *e.g.* $16 \times 16$ latent grids of 256 latents [16]. This does not directly scale to 3D shapes, as a $16 \times 16 \times 16$ grid is too large to be comfortably trained on 4-8 GPUs due to the quadratic complexity of the transformer architecture [55]. As a result, generative models based on low-resolution regular grids lead to artifacts in the details, whereas our representation yields a nice and clean surface (*e.g.* Fig. 9).

We show the following applications of our proposed representations (see Fig. 1). For 3D shape reconstruction, we show results for 3D reconstruction from a point cloud. For generative modeling, we show results for image-conditioned 3D shape generation, object category-conditioned 3D shape generation, and point-cloud-conditioned 3D shape generation. The generative model is probabilistic and can generate multiple different 3D shapes for the same conditioning information.

We summarize the contributions of our work as follows: 1) We propose irregular latent grids as 3D shape representation for neural fields. Our shape representation thereby extends existing works using a global latent, a local latent grid or a multi-scale grid. 2) We improve upon grid-based SOTA methods for 3D shape reconstruction for point-clouds. 3) We improve state-of-the art generative models for 3D shape generation, including image-conditioned generation, object category-conditioned generation and point cloud-conditioned generation.

## 2 Related Work

### 2.1 Neural shape representations

Shape analysis with neural networks processes shapes in different representations. Common representations include voxels [27, 10, 12, 44] and point clouds [40, 41, 57, 58, 51]. More recently, researchers study shapes represented with neural fields [63], *e.g.*, signed distance functions (SDFs) or occupancy (indicator) functions of shapes modeled by neural networks. Subsequently, meshes can be extracted by contouring methods such as marching cubes [28]. The methods have been called

*neural implicit representations* [29, 30, 36, 18, 13, 7, 66] or coordinate-based representations [47]. We decided to use the term *neural fields* in this paper [63].

**Point cloud processing.** Earlier works for point cloud processing rely on multilayer perceptrons (MLPs), *e.g.*, PointNet [40]. PointNet++ [41] and DGCNN [57] extended the idea by employing a hierarchical structure to capture local information. Inspired by ViT [15], which treats images as a set of patches, PCT [20] and PT [67] propose a transformer backbone for point cloud processing. Both works introduce extra modules which are not standard transformers [55] anymore. Since our main goal is not to develop a general backbone for point clouds, we simply work with a *standard* transformer for shape autoencoding (first-stage training). Similarly, both PointBERT[65] and PointMAE [34] use standard transformers for point cloud self-supervised learning.

**Neural fields for 3D shapes.** One possible approach is to represent a shape with a single *global* latent, *e.g.*, OccNet [29], CvxNet [13] and Zhang *et al.* [66]. While this method is very simple, it is not suitable to encode surface details or shapes with complex topology. Later works studied the use of *local* latents. Compared to global latents, these representations represent shapes with multiple local latents. Each latent is responsible for reconstruction in a local region. ConvONet [39], as a follow-up work of OccNet, learns latents on a grid (2d or 3d). IF-Net [9], trains a model which outputs 3d latent grids of several different resolutions. The grids are then concatenated into a final representation. Some more recent works [17, 26, 49] learn to put latents on 3D positions.

## 2.2 Generative models

Generative models include generative adversarial networks (GANs) [19], variational autoencoders (VAEs) [24], energy-based models [25, 59, 61, 62, 60], normalizing flows (NFs) [43, 14] and auto-regressive models (ARs) [54]. Our shape representation is designed to be compatible with auto-regressive generative models. Thus we will mainly discuss related work in this area.

In the image domain, earlier AR works generate pixels one-by-one, *e.g.*, PixelRNN [54] and its follow-up works [52, 45]. Combining this idea with autoregressive transformers, similar approaches are applied to 3D data, *e.g.*, PointGrow [46] for point cloud generation and PolyGen [33] for polygonal mesh generation. Other use cases include floor plan generation [35] and indoor scene generation [56, 38].

An important ingredient of many auto-regressive models is vector quantization (VQ) that was originally integrated into VAEs. VQVAE [53] adopts a two-stage training strategy. The first stage is an autoencoder with vector quantization, where bottleneck latents are quantized to discrete indices. In the second stage, an autoregressive model is trained to predict the discrete indices. This idea is further developed in VQGAN [16] which builds the autoregressive model with a transformer [55]. Vector quantization, has been shown to be an efficient way for generating high resolution images. We study how VQ can be used in the 3D domain, specifically, *neural fields* for 3D shapes.

There are some works adopting VQVAE to the 3D domain. Canonical Mapping [8] proposes point cloud generation models with VQVAE. In concurrent work, AutoSDF [32] generalizes VQVAE to 3D voxels. However, the bottleneck latent grid resolution is low and it is difficult to capture details as in ConvONet and IF-Net. Increasing the resolution is the key for detailed 3D shape reconstruction. On the other hand, higher resolution leads to more difficult training of autoregressive transformers. Another concurrent work, ShapeFormer [64], shows a solution by sparsifying latent grids. However, ShapeFormer still only accepts voxel grids as input (point clouds need to be voxelized). Additionally, the sequence length varies for different shapes, requiring the method to go through a whole dataset to find a maximum sequence length. This relies on dataset-dependent maximum sequence lengths. In contrast to existing works, our method processes point clouds directly, learns fixed-size latents and outputs neural fields. A comparison summary can be found in Table 1.

## 3 Shape Reconstruction

Here we build a shape autoencoding method. Given a shape surface, we want to output the same shape surface. We preprocess a shape surface via uniform sampling to a point cloud $\{\mathbf{x}_i : \mathbf{x}_i \in \mathbb{R}^3\}_{i \in \mathcal{N}}$ (also in matrix form $\mathbf{X} \in \mathbb{R}^{N \times 3}$), our goal is to predict the indicator function $\mathcal{O}(\cdot) : \mathbb{R}^3 \to [0, 1]$ corresponding to the volume represented by the point cloud. Our representation is being generated in

Table 1: **Method comparison.** All methods here take point clouds as input. If the column **Requiring Voxelization** says yes, it means the method takes voxels as input. The numbers shown in parenthesis are alternative choices of hyperparameters.

| | Local Latents | Requiring Voxelization | Output | Latent Resolution | Vector Quantization | Sequence Length | Comments |
|---|---|---|---|---|---|---|---|
| OccNet [29] | ✗ | ✗ | Fields | 1 | ✗ | 1 | |
| ConvONet [39] | ✓ | ✓ | Fields | $32^3(64^3)$ | ✗ | $32^3(64^3)$ | |
| IF-Net [9] | ✓ | ✓ | Fields | $128^3$ | ✗ | $128^3$ | Multiscale |
| AutoSDF [32] | ✓ | ✓ | Voxels | $8^3$ | ✓ | $8^3$ | |
| CanMap[8] | ✓ | ✗ | Point Cloud | 128 | ✓ | 128 | |
| ShapeFormer [64] | ✓ | ✓ | Fields | $16^3(32^3)$ | ✓ | variable | Sparsification |
| 3DILG (Ours) | ✓ | ✗ | Fields | 512 | ✓ | 512 | |

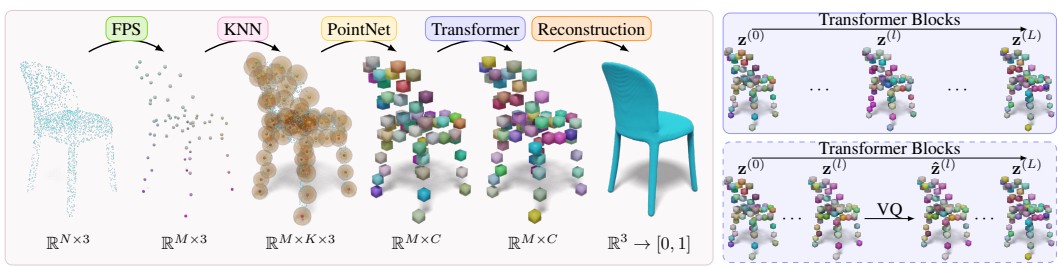

Figure 3: **Shape reconstruction from point clouds. Left:** the main pipeline. The framework can be used with (**Bottom Right**) or without (**Top Right**) vector quantization.

three steps (See Fig. 3): patch construction (Sec. 3.1), patch information processing (Sec. 3.2), and reconstruction (Sec. 3.3).

## 3.1 Patch Construction

We sub-sample the input point cloud via Farthest Point Sampling (FPS),

$$\boxed{\text{FPS}}\left(\{\mathbf{x}_i\}_{i\in\mathcal{N}}\right) = \{\mathbf{x}_i\}_{i\in\mathcal{M}} = \mathbf{X}_\mathcal{M} \in \mathbb{R}^{M\times 3}, \tag{1}$$

where $\mathcal{M} \subset \mathcal{N}$ and $|\mathcal{M}| = M$. Next, for each point in the sub-sampled point set $\{\mathbf{x}_i\}_{i\in\mathcal{M}}$, we apply the K-nearest neighbor (KNN) algorithm to find $K-1$ points to form a point patch of size $K$,

$$\forall i \in \mathcal{M}, \quad \boxed{\text{KNN}}(\mathbf{x}_i) = \{\mathbf{x}_j\}_{j\in\mathcal{N}_i}, \tag{2}$$

where $\mathcal{N}_i$ is the neighbor index set for point $\mathbf{x}_i$ and $|\mathcal{N}_i| = K-1$. Thus we have a collection of point patches,

$$\{(\mathbf{x}_i, \{\mathbf{x}_j\}_{j\in\mathcal{N}_i}) : |\mathcal{N}_i| = K-1\}_{i\in\mathcal{M}} = \mathbf{X}_{\mathcal{M},K} \in \mathbb{R}^{M\times K\times 3}. \tag{3}$$

We project each patch with a mini-PointNet-like [40] module to an embedding vector

$$\forall i \in \mathcal{M}, \quad \boxed{\text{PointNet}}(\mathbf{x}_i, \{\mathbf{x}_j\}_{j\in\mathcal{N}_i}) = \mathbf{e}_i \in \mathbb{R}^C, \tag{4}$$

where $C$ is the embedding dimension for patches.

## 3.2 Transformer

Furthermore, we build a transformer to learn local latents $\{\mathbf{z}_i\}_{i\in\mathcal{M}}$. The point coordinates $\{\mathbf{x}_i\}_{i\in\mathcal{M}}$ are converted to positional embeddings (PEs) [31], $\mathbf{p}_i = \text{PE}(\mathbf{x}_i) \in \mathbb{R}^C$,

$$\text{PE}(\mathbf{x}) = [\sin(2^0\mathbf{x}), \cos(2^0\mathbf{x}), \sin(2^1\mathbf{x}), \cos(2^1\mathbf{x}), \cdots, \sin(2^7\mathbf{x}), \cos(2^7\mathbf{x})]. \tag{5}$$

Our transformer takes as input the sequence of patch-PE pairs,

$$\boxed{\text{Transformer}}\left(\{(\mathbf{e}_i, \mathbf{p}_i)\}_{i\in\mathcal{M}}\right) = \{\mathbf{z}_i : \mathbf{z}_i \in \mathbb{R}^C\}_{i\in\mathcal{M}}. \tag{6}$$

The transformer is composed of $L$ blocks. We denote the intermediate outputs of all blocks as $\mathbf{z}_i^{(0)}, \mathbf{z}_i^{(1)}, \cdots, \mathbf{z}_i^{(l)}, \cdots, \mathbf{z}_i^{(L)}$, where $\mathbf{z}_i^{(0)}$ is the input $\mathbf{e}_i$ and $\mathbf{z}_i^{(L)}$ is the output $\mathbf{z}_i$.

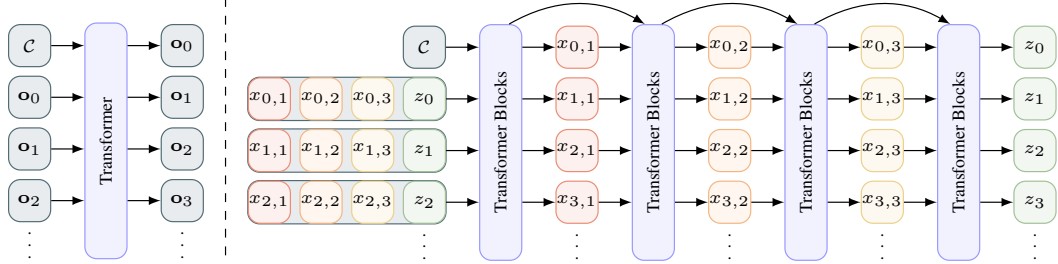

Figure 4: **Autoregressive Generative Models with Unidirectional Transformer. Left**: sequence prediction. **Right**: detailed visualizations with sequence element components.

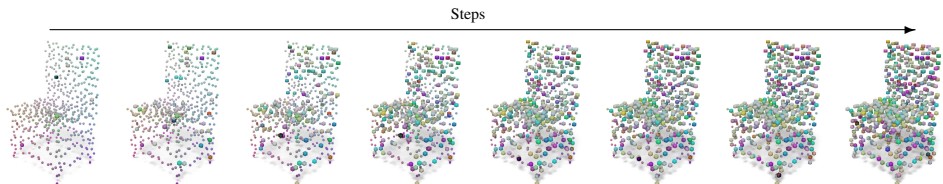

Figure 5: **Autoregressive Generative Models with Bidirectional Transformer. Cubes** ($\square$) are predicted tokens. From left to right, we show 8 decoding steps.

**Vector Quantization.** This model can be used with vector quantization [53]. We simply replace the intermediate output at block $l$ with its closest vector in a dictionary $\mathcal{D}$,

$$\forall i \in \mathcal{M}, \quad \underset{\hat{\mathbf{z}}_i^{(l)} \in \mathcal{D}}{\arg\min} \left\| \hat{\mathbf{z}}_i^{(l)} - \mathbf{z}_i^{(l)} \right\|. \tag{7}$$

The dictionary $\mathcal{D}$ contains $|\mathcal{D}| = D$ vectors. As in VQVAE [53], the gradient with respect to $\mathbf{z}_i^{(l)}$ is approximated with the straight-through estimator [1].

### 3.3 Reconstruction

We expect that each latent $\mathbf{z}_i$ is responsible for shape reconstruction near the point $\mathbf{x}_i$. However, our goal is to estimate $\mathcal{O}(\cdot)$ for an arbitrary $\mathbf{x} \in \mathbb{R}^3$. We interpolate latent $\mathbf{z}_\mathbf{x}$ for point $\mathbf{x}$ with the Nadaraya-Watson estimator,

$$\mathbf{z}_\mathbf{x} = \frac{\sum_{i \in \mathcal{M}} \exp(-\beta \|\mathbf{x} - \mathbf{x}_i\|^2) \mathbf{z}_i}{\sum_{i \in \mathcal{M}} \exp(-\beta \|\mathbf{x} - \mathbf{x}_i\|^2)}, \tag{8}$$

where $\beta$ controls the smoothness of interpolation and can be fixed or learned. The final indicator is an MLP with a sigmoid activation,

$$\hat{\mathcal{O}}(\mathbf{x}) = \text{Sigmoid}(\text{MLP}(\mathbf{x}, \mathbf{z}_\mathbf{x})). \tag{9}$$

**Loss Function.** We optimize the estimated $\hat{\mathcal{O}}(\cdot)$ by comparing to ground-truth $\mathcal{O}(\cdot)$ via a binary cross entropy loss (BCE) and an additional commitment loss term [53],

$$\mathcal{L} = \mathcal{L}_{\text{recon}} + \lambda \mathcal{L}_{\text{commit}} = \mathbb{E}_{\mathbf{x} \in \mathbb{R}^3} \left[ \text{BCE}(\hat{\mathcal{O}}(\mathbf{x}), \mathcal{O}(\mathbf{x})) \right] + \lambda \mathbb{E}_{\mathbf{x} \in \mathbb{R}^3} \left[ \mathbb{E}_{i \in \mathcal{M}} \left\| \text{sg}(\hat{\mathbf{z}}_i^{(l)}) - \mathbf{z}_i^{(l)} \right\|^2 \right], \tag{10}$$

where $\text{sg}(\cdot)$ is the stop-gradient operation. When $\lambda = 0$, it means we train the model without vector quantization.

## 4 Autoregressive Generative Modeling

With Vector Quantization (Eq. 7), we compress the bit size of intermediate latents $\{\hat{\mathbf{z}}_i^{(l)}\}_{i \in \mathcal{M}}$ to $\log_2 D$ where $D$ is the size of the dictionary $\mathcal{D}$. We denote the compressed index as $z_i \in \{0, 1, \ldots, D-1\}$. We also quantize point coordinates $\mathbf{x}_i$ to $(x_{i,1}, x_{i,2}, x_{i,3})$ where each entry is a 8-bit integer $\{0, 1, \ldots, 255\}$, As a result, we obtain a discrete representation of the 3D shape $\{(x_{i,1}, x_{i,2}, x_{i,3}, z_i)\}_{i \in \mathcal{M}}$.

### 4.1 Unidirectional Transformer

Autoregressive generation with unidirectional transformer is a more classical approach. We often need to sequentialize an unordered set if the order is not defined yet. Specifically, we re-order the representations in ascending order by the first component $x_{i,1}$, then by the second component $x_{i,2}$, and finally by the third component $x_{i,3}$,

$$\mathcal{S} = \{(x_{0,1}, x_{0,2}, x_{0,3}, z_0), (x_{1,1}, x_{1,2}, x_{1,3}, z_1), \ldots, (x_{M-1,1}, x_{M-1,2}, x_{M-1,3}, z_{M-1})\}. \quad (11)$$

Our goal to predict sequence elements one-by-one. A common way is to flatten the sequence $\mathcal{S}$ by concatenating the quadruplets, for examples, PointGrow [46] and PolyGen [33]. However, here we consider an approach generating *quadruplet-by-quadruplet*. We write $\mathbf{o}_i = (x_{i,1}, x_{i,2}, x_{i,3}, z_i)$. The likelihood of generating $\mathbf{o}_i$ autoregressively is

$$p(\mathcal{S} \mid \mathcal{C}) = \prod_{i=0}^{i=M-1} p(\mathbf{o}_i \mid \mathbf{o}_{<i}, \mathcal{C}), \quad (12)$$

where $\mathcal{C}$ is conditioned context. Similar to ATISS [38], we predict components of $\mathbf{o}_i = (x_{i,1}, x_{i,2}, x_{i,3}, z_i)$ autoregressively:

$$\begin{aligned} &p(\mathbf{o}_i \mid \mathbf{o}_{<i}, \mathcal{C}) \\ &= p(x_{i,1} \mid \mathbf{o}_{<i}, \mathcal{C}) \cdot p(x_{i,2} \mid x_{i,1}, \mathbf{o}_{<i}, \mathcal{C}) \cdot p(x_{i,3} \mid x_{i,2}, x_{i,1}, \mathbf{o}_{<i}, \mathcal{C}) \cdot p(z_i \mid x_{i,3}, x_{i,2}, x_{i,1}, \mathbf{o}_{<i}, \mathcal{C}). \end{aligned} \quad (13)$$

The model is shown in Fig. 4. Different from ATISS [38] which uses MLPs to decode different components, instead we continue to apply transformer blocks for component decoding.

### 4.2 Bidirectional Transformer

Bidirectional transformer for autoregressive generation is recently proposed by MaskGIT [5]. Here we show our model can also be combined with bidirectional transformer. However, here we consider a different task. We generate $\{z_i\}_{i\in\mathcal{M}}$ conditioned on $\{(x_{i,1}, x_{i,2}, x_{i,3})\}_{i\in\mathcal{M}}$. In training phase, we sample a subset $\{z_i\}_{i\in\mathcal{V}\subset\mathcal{M}}$ of $\{z_i\}_{i\in\mathcal{M}}$ as the input of the bidirectional transformer. We aim to predict $\{z_i\}_{i\in\mathcal{M}\setminus\mathcal{V}}$. The coordinates $\{(x_{i,1}, x_{i,2}, x_{i,3})\}_{i\in\mathcal{M}}$ are converted to positional embeddings (either learned or fixed) as condition. The likelihood of generating $\mathcal{M} \setminus \mathcal{V}$ is as follows,

$$\prod_{\mathcal{V}\subset\mathcal{M}} p(\{z_i\}_{i\in\mathcal{M}\setminus\mathcal{V}} \mid \{z_i\}_{i\in\mathcal{V}}, \{(x_{i,1}, x_{i,2}, x_{i,3})\}_{i\in\mathcal{M}}). \quad (14)$$

In practice, the bidirectional transformer takes as input all tokens except that $\{z_i\}_{i\in\mathcal{M}\setminus\mathcal{V}}$ are replaced by a special masked token $[m]$. When decoding (inference), we iteratively predict multiple tokens at the same time. Tokens are sampled based on their probabilities (transformer output). See [5] for a detailed explanation. We show visualization of decoding steps in Fig. 5.

## 5 Reconstruction Experiments

We set the size of the input point cloud to $N = 2048$. The number and the size of point patches are $M = 512$ and $K = 32$, respectively. In the case of Vector Quantization, there are $D = 1024$ vectors in the dictionary $\mathcal{D}$. Other details of the implementation can be found in the Appendix. We use the datset ShapeNet-v2 [4] for shape reconstruction. We split samples into train/val/test (48597/1283/2592) set. Following the evaluation protocol of [13, 66], we include three metrics, volumetric IoU, the Chamfer-L1 distance, and F-Score [50]. We also show reconstruction results on another object-level dataset ABO [11], a real-world dataset D-FAUST [2], and a synthetic scene-level dataset in the appendix.

**Results**  We compare our method with three existing works, OccNet [29], ConvONet [39] and IF-Net [9]. The results can be found in Table 2. We select 7 categories among 55 categories with largest training set (*table, car, chair, airplane, sofa, rifle, lamp*). Detailed results can be found in the Appendix. We show different choices of $M$. As we can see in this table, increasing $M$ from 64 to 512 gives a performance boost. Even with the simplest model $M = 64$, our results outperform ConvONet. The best results are achieved when setting $M = 512$. In this case, our results lead in most categories

Table 2: **Shape reconstruction.** We train all models on ShapeNet-v2 (55 categories). All baseline methods are trained with the corresponding officially released code. For our model, we set $N = 2048$ and $K = 32$. We select 7 categories to show. These categories have largest training set among 55 categories. We also show averaged metrics over all categories. The numbers shown in parenthesis are results of vector quantization.

| | | OccNet | ConvONet | IF-Net | 3DILG (Ours) | | | |
| --- | --- | --- | --- | --- | --- | --- | --- | --- |
| | | | | | $M = 64$ | $M = 128$ | $M = 256$ | $M = 512$ |
| IoU ↑ | mean (selected) | 0.822 | 0.881 | 0.929 | 0.922(0.904) | 0.936(0.929) | 0.945(0.943) | **0.952**(0.950) |
| | mean (all) | 0.825 | 0.888 | 0.934 | 0.923(0.907) | 0.937(0.929) | 0.946(0.943) | **0.953**(0.950) |
| Chamfer ↓ | mean (selected) | 0.058 | 0.040 | 0.034 | 0.038(0.038) | 0.035(0.036) | 0.034(0.034) | **0.032**(0.030) |
| | mean (all) | 0.072 | 0.052 | 0.041 | 0.048(0.052) | 0.044(0.046) | 0.041(0.042) | **0.040**(0.040) |
| F-Score ↑ | mean (selected) | 0.898 | 0.951 | 0.975 | 0.959(0.948) | 0.968(0.964) | 0.972(0.969) | **0.976**(0.975) |
| | mean (all) | 0.858 | 0.933 | **0.967** | 0.942(0.926) | 0.955(0.948) | 0.963(0.958) | 0.966(0.965) |

when comparing to IF-Net. The results of IF-Net are better than ours in terms of metric F-Score in some categories. However, we argue that in this case the metric F-score is saturated (values are close to 1), which making it hard to compare. We also show results after introducing vector quantization to our model in the same table. We can see that vector quantization harms the performance slightly.

**Qualitative Comparison**    Qualitative results are shown in Fig. 6. From the visualization, it can be seen that the reconstruction quality increases as $M$ increases, particularly for shapes with complex topology. OccNet, as a global latent method, often fails to recover complex structures. ConvONet can recover better structures than OccNet, due to its localized latents. By learning multiscale latents, IF-Net improves further upon ConvONet. Our method with $M = 64$ outperforms ConvONet, and with $M = 128$, the results are comparable with IF-Net.

## 6   Generative Experiments

We introduce three experiments to show how our model can be combined with auto-regressive transformers for generative modeling. In Sec. 6.1, we show image-conditioned generation as probabilistic shape reconstruction from single image. In Sec. 6.2, we show how we generate samples given a category label (using the 55 ShapeNet categories). In Sec. 6.3, we further show generation conditioned on downsampled point clouds. In contrast to the first two tasks, the point-cloud-conditioned generation is based on a bidirectional transformer described in Sec 4.2. More generative results on additional datasets ABO and D-FAUST can be found in the appendix.

### 6.1   Probabilistic Shape Reconstruction from a Single Image

We train a uni-directional transformer for this task (see Sec. 4.1). The context $\mathcal{C}$ is an image. To train this model, we render 40 images ($224 \times 224$) of different views for each shape in ShapeNet. The implementation of the uni-directional transformer is based on GPT [42]. It contains 24 blocks, where each block has an attention layer with 16 heads and 1024 embedding dimension. When sampling, nucleus sampling [22] with top-$p$ (0.85) and top-$k$ (100) are applied to predicted token probabilities.

**High resolution images.**    We show some generated samples when $\mathcal{C}$ are high resolution images in Fig. 7. We compare our results with a deterministic method, OccNet [29]. As we can see in the results, the deterministic method (OccNet) tends to create blurred meshes. However, our probabilistic reconstruction is able to output detailed meshes.

**Low resolution images.**    We also consider another more challenging task. The input images are downsampled to low resolution ($16 \times 16$). In this case, the generative model has more freedom to find multiple plausible interpretations, including variations with different topology (see Fig. 8).

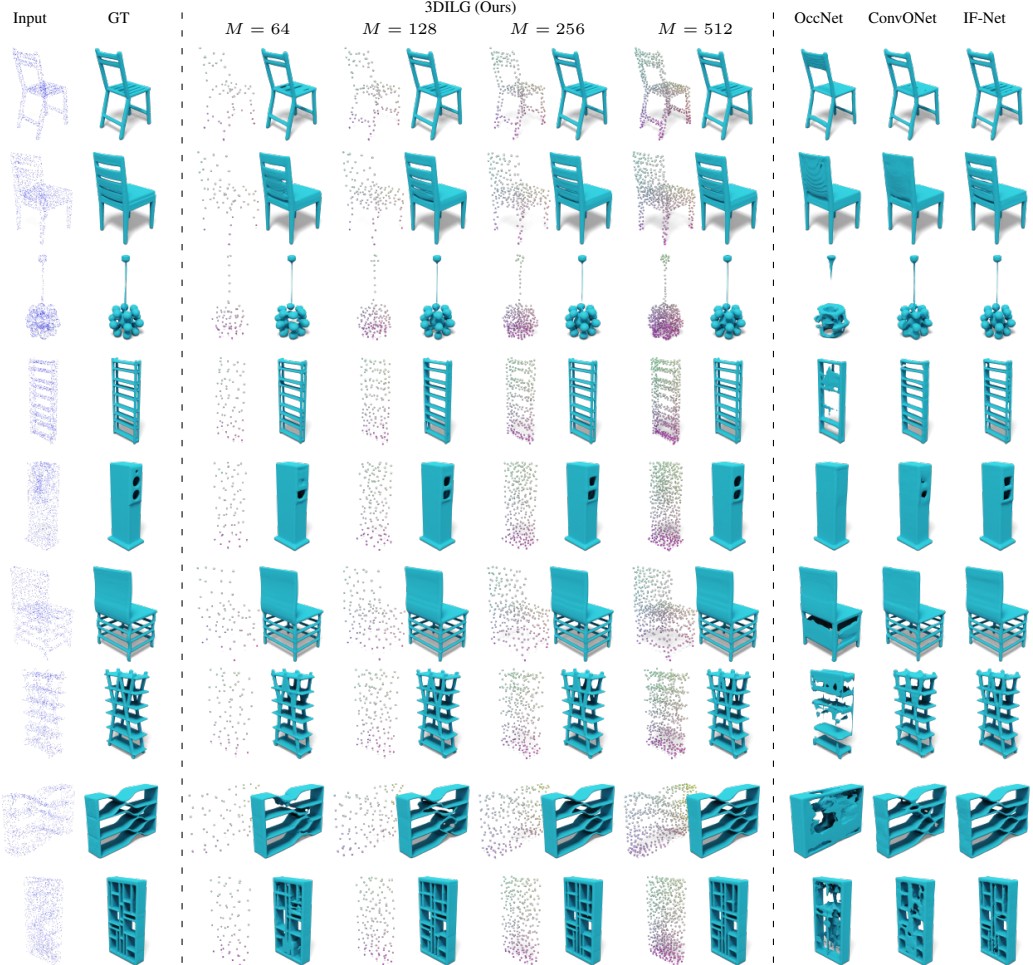

Figure 6: **Shape reconstruction.** The column **Input** shows input point clouds of size 2048. The column **GT** shows ground-truth meshes. We compare our results with different $M$ to OccNet, ConvONet and IF-Net. We also show $\{\mathbf{x}_i\}_{i \in \mathcal{M}}$ obtained via Farthest Point Sampling.

## 6.2 Category-Conditioned Generation

To generate shapes given a category label, we use the context $\mathcal{C}$ to encode the category label. We employ the uni-directional transformer based on GPT as in Sec. 6.1. We compare to our proposed baseline model that encodes shapes as a latent grid of resolution $8^3$. To make a fair comparison, we also extend ViT [15] to the 3D voxel domain in the first stage of training. In the second stage, we use the same uni-directional transformer for sequence prediction. Here the sequence length is $8^3 = 512$, which is the same as in our proposed model. The baseline model is named **Grid-**$8^3$. The comparison to the baseline model is in Fig. 9. We can see that the baseline is unable to generate high quality shapes. We argue that this is because the representation is not expressive enough to capture surface details. Furthermore, we show more generated samples of our model in Fig. 10. The three selected categories are *bookshelf, bench* and *chair*. For both the baseline and our model, we render 10 images of predicted shapes, and calculate the Fréchet Inception Distance (FID) [21, 37] between predictions and test sets. The metrics can be found in Table 3. A perceptual study on the quality of generated samples can be found in the appendix.

## 6.3 Point Cloud Conditioned Generation

We train a bidirectional transformer described in Sec 4.2. The model takes as input $\{\mathbf{x}_i\}_{i \in \mathcal{M}}$. The results can be found in Fig. 11.

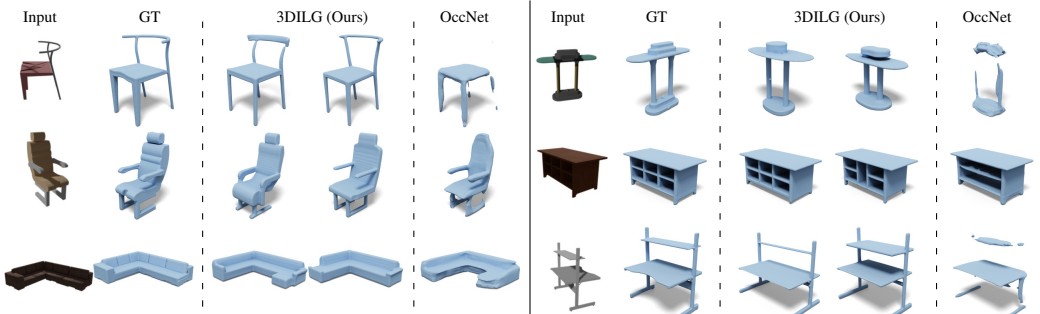

Figure 7: **Image-conditioned generation** ($224 \times 224$). We sample 2 shapes for each input image, and compare them with OccNet.

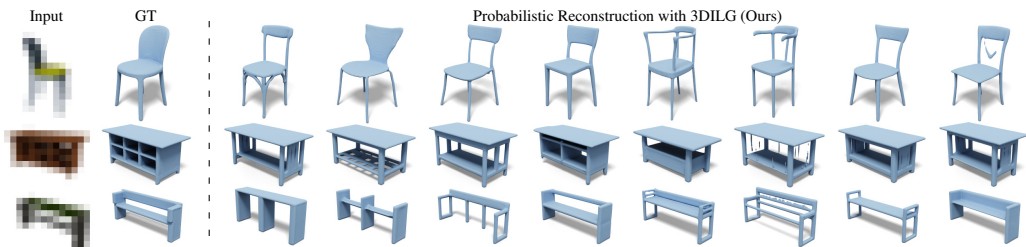

Figure 8: **Image-conditioned generation** ($16 \times 16$). We sample 8 shapes for each input image.

## 7 Conclusion

We have studied neural fields for shapes and presented a new representation. In contrast to common approaches which define latents a on a regular grid or multiple regular grids, we position latents on an irregular grid. Comparing to existing works, the representation better scales to larger models, because the irregular grid is sparse and adapts to the underlying 3D shape of the object. In our results, we demonstrated an improvement in 3D shape reconstruction from point clouds and generative modeling conditioned on images, object category, or point-clouds over alternative grid-based methods. In future work, we suggest to explore other applications of our proposed representation, *e.g.*, shape completion, and extensions to textured 3D shapes and 3D scenes.

## Broader impact

We introduce a new 3d shape representation for generative modeling and shape analysis. This shape representation is designed to be compatible with the transformer architecture. We demonstrate some example applications in the paper including shape generation conditioned on images, point clouds, and shape class, and 3d shape reconstruction. However, we envision our proposed shape representation to be general and it could be employed in all shape processing tasks.

Potential societal impacts of generative modeling in general exist. Future iterations of our work could possibly be used to generate high fidelity 3D virtual humans. However, we do not see an important negative societal impact that is specific to our work and that would constitute an immediate concern.

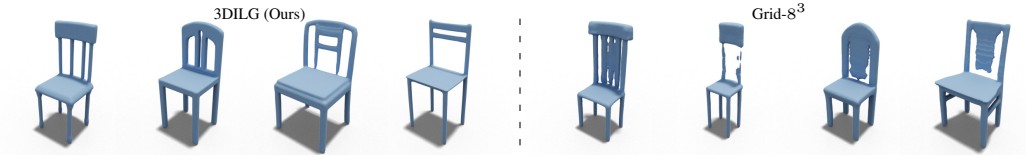

Figure 9: **Comparison of category-conditioned generation.** We compare our results (**Left**) with an $8^3$ latent grid baseline (**Right**).

Table 3: **FID ↓ for category-conditioned generation**. We compare our results with the baseline Grid-8³. The 7 categories are the largest categories in ShapeNet.

| | Categories | | | | | | | mean |
|---|---|---|---|---|---|---|---|---|
| | table | car | chair | airplane | sofa | rifle | lamp | |
| Grid-8³ | 72.396 | 95.566 | 58.649 | 42.009 | 58.092 | 59.456 | 87.319 | 67.641 |
| 3DILG (Ours) | **68.016** | **92.597** | **45.333** | **30.957** | **53.244** | **40.500** | **72.672** | **57.617** |

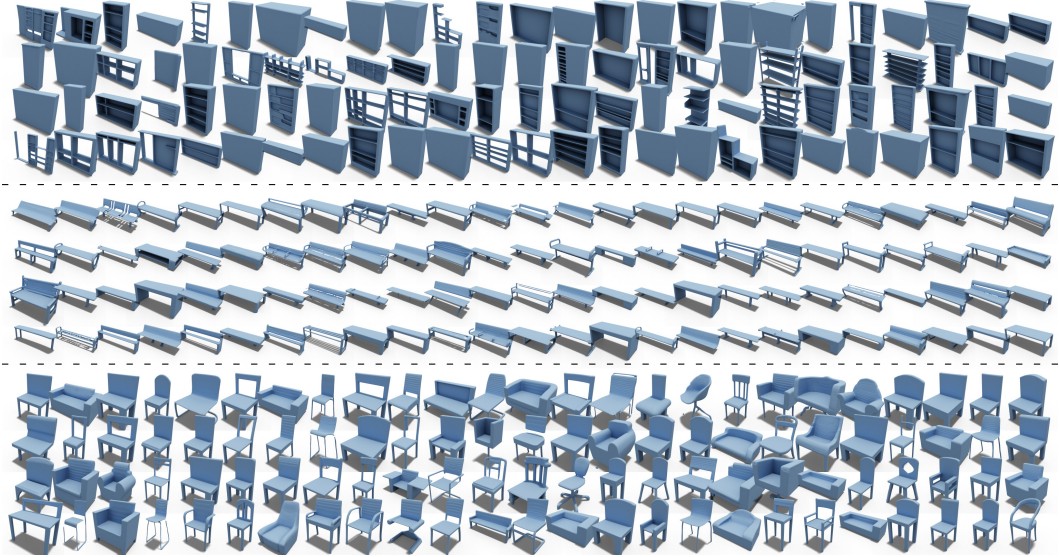

Figure 10: **Category-conditioned generation.** We choose 3 categories to show (*bookshelf, bench* and *chair*). We show 100 samples for each category.

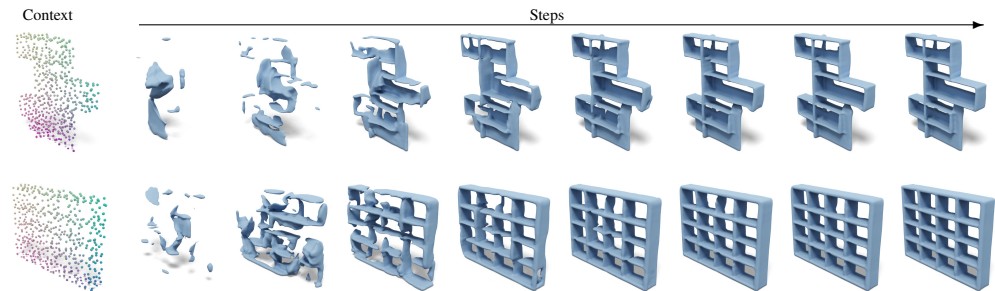

Figure 11: **Point cloud conditioned generation.** We show 8 decoding steps.

## Acknowledgements

We would like to acknowledge support from the SDAIA-KAUST Center of Excellence in Data Science and Artificial Intelligence.

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
