# OpenReview forum: "3DILG: Irregular Latent Grids for 3D Generative Modeling"
_NeurIPS.cc/2022/Conference — NeurIPS 2022 Accept_

### Official Review · Reviewer_uMXb · 2022-07-11

**Rating:** 7
**Confidence:** 4
**Soundness:** 4 excellent
**Presentation:** 3 good
**Contribution:** 3 good

**Summary:**

This submission proposes to irregularly distribute local SDF latent codes in 3D, to adapt to shapes and not be restricted to a 3D grid with fixed resolution.
In addition, authors present ways to quantize their representation, and use it in the generative setting with uni- and bi-directional transformers.

**Questions:**

l. 33 : Why is fixed length representation a benefit? Wouldn't an adaptive representation be more useful, with a tiny representation for simple shapes, and long representation for complex ones?

l.141 What is the motivation behind the non linear interpolation between local latent codes? Could it be ablated?

**Limitations:**

Compared to autodecoders, this probabilistic pipeline can no longer represent a shape with a latent code.

**Strengths And Weaknesses:**

Strengths:
- excellent technical presentation. The use of colors in the text and equations to refer to different pipeline parts is a great idea.
- generating multiple plausible outputs given 1 single conditioning signal (eg. Fig. 10) is a very good use of probabilistic models
- results look qualitatively good, and the proposed pipeline is properly compared against recent models (IF-Net, ConvOccNet) that distribute latent codes on fixed grids.


Weaknesses:
- Quantizing 3D positions of local latents sounds very similar to using a grid. What is the difference here?
- Some very related papers such as "Local Deep Implicit Functions for 3D Shape" [CVPR 2020] are not mentioned. In this one, "the global implicit function is decomposed into the sum of N local implicit functions.", not on a grid...
- The intro+related Sections mention shape representation and generation, while the method section starts with an input point cloud.

---

> ### Author Response · Authors · 2022-08-02
> **Response to Reviewer uMXb Part 1**
>
> > Quantizing 3D positions of local latents sounds very similar to using a grid. What is the difference here?
>
> We agree with this as we can easily predict continuous point coordinates, thus making the quantization not necessary. However, some works found that quantization leads to high accuracy, e.g., PolyGen [1] and DORN [2]. The quantization converts a regression problem to a classification problem. We conjecture that a classification problem might be easier than a regression problem for deep networks. But digging into the reason behind this is beyond the scope of our paper. We follow (current) best practices suggested by previous papers [1] [2].
>
> On the other hand, we are using 8-bit quantization which leads to a 256x256x256 grid. In this sense, the resolution is larger than all previous works, according to Table 1. Not to mention, we can still increase the bit length, e.g., increase to 14-bit. In this sense, we have a higher precision in localizing latents than previous work using only a grid.
>
> > Some very related papers such as "Local Deep Implicit Functions for 3D Shape" [CVPR 2020] are not mentioned. In this one, "the global implicit function is decomposed into the sum of N local implicit functions.", not on a grid...
>
> We have added the missing reference.
>
> > The intro+related Sections mention shape representation and generation, while the method section starts with an input point cloud.
>
> Thanks for pointing this out. Sampling point clouds from surfaces is a data preprocessing step. We did as suggested to mention surfaces at the beginning of this section as the input of our method.
>
> > l. 33 : Why is fixed length representation a benefit? Wouldn't an adaptive representation be more useful, with a tiny representation for simple shapes, and long representation for complex ones?
>
> We consider the fixed length representation to be a trade off with distinct benefits in our application. If the representation is variable length, we need to go through all data samples in the dataset to find a maximum length. Also, in training, we have to pad shorter sequences to the largest length in the data batch, just as what people do in natural language processing. It’s more convenient to deal with fixed length representations in auto-regressive models or models using masked tokens.

---

> > ### Author Response · Authors · 2022-08-02
> > **Response to Reviewer uMXb Part 2**
> >
> > > l.141 What is the motivation behind the non linear interpolation between local latent codes? Could it be ablated?
> >
> > Nadaraya–Watson estimator is a non-parametric regression estimator. We choose it for the following reasons:
> >
> > 1.  It is widely used in machine learning ([4] Sec 14.7.4)
> >
> > 2.  It can be easily implemented with the softmax function (most deep learning frameworks provided a numerically stable version of softmax)
> >
> > 3.  The attention operator used in Transformers can be seen as a special case of the Nadaraya-Watson estimator [5]. From this perspective, we allow any point to attend all learned latent locations.
> >
> > 4.  Most importantly, it is efficient to backpropagate through the interpolation to design an end to end learning framework.
> >
> >
> > There are many other interpolation methods that exist, e.g., kriging in geo-statistics. However, making them differentiable is not that straightforward. We do agree that changing the interpolation might even improve our method further so we follow the reviewer suggestion. We show results of another simple interpolation methods which only considers k nearest neighbors for the interpolation. It is called knn interpolation in PointNet++ [3]. As shown in the table, the results are very close (improved 0.004). The network can automatically learn latents that are suitable for the interpolation method. So we believe that the interpolation method will be a minor issue in our work and orthogonal to our contribution. However, there is definitely some potential for future work.
> > |       | Nadaraya-Watson |  KNN  | Improvements |
> > |-------|:---------------:|:-----:|:------------:|
> > | IoU ↑ |      0.953      | 0.957 |     0.004    |
> >
> > > Compared to autodecoders, this probabilistic pipeline can no longer represent a shape with a latent code.
> >
> > However, we can represent a shape with a set of latent codes. The set is much smaller compared to ConvOccNet and IF-Net. Also an autoencoder may use a latent code with many dimensions.
> >
> >
> >
> > ### References
> >
> > [1] Nash, Charlie, Yaroslav Ganin, SM Ali Eslami, and Peter Battaglia. "Polygen: An autoregressive generative model of 3d meshes." In International conference on machine learning, pp. 7220-7229. PMLR, 2020.
> >
> > [2] Fu, Huan, Mingming Gong, Chaohui Wang, Kayhan Batmanghelich, and Dacheng Tao. "Deep ordinal regression network for monocular depth estimation." In Proceedings of the IEEE conference on computer vision and pattern recognition, pp. 2002-2011. 2018.
> >
> > [3] Qi, Charles Ruizhongtai, Li Yi, Hao Su, and Leonidas J. Guibas. "Pointnet++: Deep hierarchical feature learning on point sets in a metric space." Advances in neural information processing systems 30 (2017).
> >
> > [4] Murphy, Kevin P. Machine learning: a probabilistic perspective. MIT press, 2012.
> >
> > [5] Zhang, Aston, Zachary C. Lipton, Mu Li, and Alexander J. Smola. "Dive into deep learning." arXiv preprint arXiv:2106.11342 (2021).

---

### Official Review · Reviewer_tLeQ · 2022-07-14

**Rating:** 5
**Confidence:** 4
**Soundness:** 3 good
**Presentation:** 3 good
**Contribution:** 3 good

**Summary:**

This paper presents a new 3D shape representation for 3D generative modeling. The proposed shape representation is formed as irregular grids of latent vectors in the 3D space that roughly forms the shape, which can be encoded with Transformer-based networks and decoded via autoregressive generation, similar to VQ-GAN. Experiments on 3D shape generation conditioned on different context is performed on ShapeNet and shows promising results.


**Questions:**

It would be great if the authors could address all the concerns listed above.

**Limitations:**

The limitations and potential societal impact have been adequately addressed.

**Strengths And Weaknesses:**

Strengths:
- The authors proposed a very interesting idea of bringing recent advances on generative image modeling into 3D shapes, specifically VQ-GAN. It nicely brings in the sparse nature of 3D shapes, and bridges together commonly accepted differentiable point cloud processing operations with autoregressive generation.
- The experimental results are pretty promising. I especially like the fact that, conditioned on a single input image, the model is capable of generating different shape modalities, which is something previous deterministic shape reconstruction methods cannot achieve.
- The paper is nicely written and the presentation is clear.

Weaknesses:
- Maybe I missed it, but it was not clear how the autoregressive generation ordering was determined (L156-158). I think the authors should also provide some discussions on whether (or how) the model is / is not invariant to the permutation of such ordering; for completeness, results comparing different input permutations may also give a better understanding.
- For the "image-conditioned generation" experiments comparing single-image shape reconstruction, I would like to understand more where the actual performance gain comes from. Besides the new irregular latent grid representation, the entire backbone architecture (GPT) is also much more powerful than OccNet (ResNet-18); even the pretraining data is entirely different. Therefore, I am not totally convinced that all the benefit comes from the new representation. I think a critical missing experiment is to ablate this factor; specifically, a baseline that uses GPT but predicting occupancy like OccNet should be included to verify that the irregular latent grids do have its merits.
- For the image-conditioned experiments (following the above concern), it was not clear whether there are still input point clouds being conditioned. If yes, then the comparison against OccNet becomes yet more unfair; if not, it is not very clear what the input pipeline would look like. My guess is that a full shape reconstruction network (Fig 3) is trained first, then only the Transformer part is taken for the experiment (with the PointNet discarded); it would be good if the authors could clarify.

---

> ### Author Response · Authors · 2022-08-02
> **Response to Reviewer tLeQ Part 1**
>
> > Maybe I missed it, but it was not clear how the autoregressive generation ordering was determined (L156-158). I think the authors should also provide some discussions on whether (or how) the model is / is not invariant to the permutation of such ordering; for completeness, results comparing different input permutations may also give a better understanding.
>
> The current permutation we are using follows related work, e.g., PointGrow [1] (Section 3) and PolyGen [2] (Section 2.2). We sort points according to their coordinates and flatten them. This is also referred to as scanline order in other papers. Such a model is not permutation invariant. We choose the ordering according to these existing works. For example, in VQGAN [3] (Fig. 47), different orderings were studied. Scan line order works best.
>
> > For the "image-conditioned generation" experiments comparing single-image shape reconstruction, I would like to understand more where the actual performance gain comes from. Besides the new irregular latent grid representation, the entire backbone architecture (GPT) is also much more powerful than OccNet (ResNet-18); even the pretraining data is entirely different. Therefore, I am not totally convinced that all the benefit comes from the new representation. I think a critical missing experiment is to ablate this factor; specifically, a baseline that uses GPT but predicting occupancy like OccNet should be included to verify that the irregular latent grids do have its merits.
>
> To make a fair comparison, we use the same encoder as in OccNet (ResNet-18). The different thing is the decoder part. OccNet is a deterministic method. Given an image feature vector, it outputs a shape surface deterministically. For our method, we can sample output shape probabilistically given the same image feature vector. GPT is an autoregressive text generation model. It is often used in VQVAE-like models for discrete token generation, e.g., VQGAN [3]. In general, we argue that probabilistic methods give results sampled from a probabilistic distribution, while deterministic method tends to produce averaged results (mean of the distribution). We would also like to highlight that the focus of our contribution is in generative modeling and OccNet is not a generative modeling method. Directly using GPT to predict occupancy at a dense resolution would be virtually impossible even with an excessively large cluster. For example, we currently predict 512 tokens. The large scale GPT3 model can process less than 5000 tokens. Since self-attention scales quadratically, we see no way to scale to 128 * 128 * 128 tokens.
>
> > For the image-conditioned experiments (following the above concern), it was not clear whether there are still input point clouds being conditioned. If yes, then the comparison against OccNet becomes yet more unfair; if not, it is not very clear what the input pipeline would look like. My guess is that a full shape reconstruction network (Fig 3) is trained first, then only the Transformer part is taken for the experiment (with the PointNet discarded); it would be good if the authors could clarify.
>
> Once the shape reconstruction network (Stage One) was trained, it can be used in all generative models (Stage Two), e.g., category-conditioned, image-conditioned and point-conditioned. This two-stage training strategy is also used in all VectorQuantization + AutoRegressive models, e.g., VQVAE[4], VQGAN [3], MaskGIT[7], Parti[8], AutoSDF [5], and ShapeFormer[6] (some is concurrent work).
>
> -   When training, we need image-surface pairs (which is the same as in OccNet). The shape reconstruction network (Fig 3) converts surface to discrete tokens.
>
> -   When doing inference, we do not need shape surfaces (so the comparison with OccNet is fair). Given sampled discrete tokens conditioned on images, we need the shape reconstruction network to decode the tokens to a final surfaces.
>
>
> We will make the process clearer in the next revision.

---

> > ### Author Response · Authors · 2022-08-02
> > **Response to Reviewer tLeQ Part 2**
> >
> > ### References
> > [1] Sun, Yongbin, Yue Wang, Ziwei Liu, Joshua Siegel, and Sanjay Sarma. "Pointgrow: Autoregressively learned point cloud generation with self-attention." In Proceedings of the IEEE/CVF Winter Conference on Applications of Computer Vision, pp. 61-70. 2020.
> >
> >
> >
> > [2] Nash, Charlie, Yaroslav Ganin, SM Ali Eslami, and Peter Battaglia. "Polygen: An autoregressive generative model of 3d meshes." In International conference on machine learning, pp. 7220-7229. PMLR, 2020.
> >
> >
> >
> > [3] Esser, Patrick, Robin Rombach, and Bjorn Ommer. "Taming transformers for high-resolution image synthesis." In Proceedings of the IEEE/CVF conference on computer vision and pattern recognition, pp. 12873-12883. 2021.
> >
> >
> >
> > [4] Van Den Oord, Aaron, and Oriol Vinyals. "Neural discrete representation learning." Advances in neural information processing systems 30 (2017).
> >
> >
> >
> > [5] Mittal, Paritosh, Yen-Chi Cheng, Maneesh Singh, and Shubham Tulsiani. "Autosdf: Shape priors for 3d completion, reconstruction and generation." In Proceedings of the IEEE/CVF Conference on Computer Vision and Pattern Recognition, pp. 306-315. 2022.
> >
> >
> >
> > [6] Yan, Xingguang, Liqiang Lin, Niloy J. Mitra, Dani Lischinski, Daniel Cohen-Or, and Hui Huang. "Shapeformer: Transformer-based shape completion via sparse representation." In Proceedings of the IEEE/CVF Conference on Computer Vision and Pattern Recognition, pp. 6239-6249. 2022.
> >
> >
> >
> > [7] Chang, Huiwen, Han Zhang, Lu Jiang, Ce Liu, and William T. Freeman. "Maskgit: Masked generative image transformer." In Proceedings of the IEEE/CVF Conference on Computer Vision and Pattern Recognition, pp. 11315-11325. 2022.
> >
> >
> >
> > [8] Yu, Jiahui, Yuanzhong Xu, Jing Yu Koh, Thang Luong, Gunjan Baid, Zirui Wang, Vijay Vasudevan et al. "Scaling Autoregressive Models for Content-Rich Text-to-Image Generation." arXiv preprint arXiv:2206.10789 (2022).

---

### Official Review · Reviewer_jKmw · 2022-07-16

**Rating:** 7
**Confidence:** 3
**Soundness:** 3 good
**Presentation:** 3 good
**Contribution:** 3 good

**Summary:**

The paper proposes a method for detailed shape reconstruction from point clouds and images, as well as shape generation. The method utilizes FPS and KNN together with a PointNet to create patch embeddings of train shapes, and then it encodes a shape indicator function using a transformer architecture. The paper demonstrates impressive shape reconstruction and generation results for all of the aforementioned tasks.

**Questions:**

Please see and address the questions raised in the previous section.

**Limitations:**

The authors need to address further a potential limitation of the method, namely, its sensitivity to changes in sampling of the input point clouds.

**Strengths And Weaknesses:**

Strengths
- The paper presents a novel method for encoding local shape information using irregular latent spaces and using transformer-based autoregressive models for shape reconstruction from point clouds and images.
- The paper presents impressive - accurate and detailed, shape generation results from high-res and blurred images, using compact shape latent spaces.
- The presentation is very clear.
- The paper presents a comprehensive set of experimental results illustrating the advantages of the proposed method and comparing it to current SOTA methods.

Weaknesses:
- A possible weakness of the proposed approach may be its sensitivity to shape sampling. E.g., how would the method work if the input point clouds were not all sampled at N = 2048 points? How were these points obtained? Would irregular sampling density affect the results of the proposed method? This possible limitation needs to be addressed in the paper, with some experimental results and a discussion.
Parts of the method's description which require further classification.
- In Figure 4, what’s the output of the model? How the point cloud, or another shape representation, is decoded from it?
- Figure 5 is inconsistent with Figure 4. Show a block diagram of the proposed architecture, as you do in Figure 4.
- In Section 4, how are x_{i,k} obtained for different i-s? What is the difference between x_{i,k} and x_{j,k} for different i, j? Please explain.
- In Section 5, further comparison with IF-NET is needed: the proposed method slightly improves over IF-NET. How does it compare to IF-NET (and other methods) in other respects - resource consumption, train and inference time, etc.?
- Explain the purpose of Eq.(8) and Eq.(9).
Comments regarding supplementary material.
- In A.4., how do the methods compare in terms of memory and runtime?
- In A.6., ll.62-63 - “Bad” in what sense? Non-realistic generations by the proposed method should be highlighted as well.
Other comments.
- In l.130, explicitly state which positional encodings are used, for completeness of the presentation.
- Introduction section needs to be revised to more clearly describe the main aspects of the proposed approach.
- A related work which should be cited: Y. Li, S. Pirk, H. Su, C. R. Qi, L. J. Guibas, FPNN: Field Probing Neural Networks for 3D Data, Neural Information Processing Systems (NIPS 2016).
- In Section 6.1, l.214, provide reference for GPT.
- Nit: In Table 1, consider writing all numbers in the same manner such that it will be clear which parameters are identical for different methods.

---

> ### Author Response · Authors · 2022-08-02
> **Response to Reviewer jKmw Part 1**
>
> > A possible weakness of the proposed approach may be its sensitivity to shape sampling. E.g., how would the method work if the input point clouds were not all sampled at N = 2048 points? How were these points obtained? Would irregular sampling density affect the results of the proposed method? This possible limitation needs to be addressed in the paper, with some experimental results and a discussion. Parts of the method's description which require further classification.
>
> > The authors need to address further a potential limitation of the method, namely, its sensitivity to changes in sampling of the input point clouds.
>
> We followed the reviewers suggestion and conducted additional experiments. Our method is actually surprisingly robust when used with different sampling patterns compared to previous work. To illustrate problems that might be caused by irregular density sampling we design a simple experiment. We randomly choose an “anchor” point on the surface. We want points near the “anchor” point to have a high probability to be sampled. The probabilities are defined by a Gaussian function exp(-beta * dist(p, anchor)). Here small beta gives rise to uniform sampling, while large beta assigns large sampling probability near the “anchor” point (in an extreme case, some areas on the surface will never be sampled from). We use this function to simulate an irregular density sampling on the surface. The results can be found in the following Table. We observed a significant drop in IF-Net’s results. On the contrary, we do not see a large performance drop in our method. Also see Appendix A.10 (Fig. 20) for reconstructed meshes.
>
> |          |         | IF-Net |        |         |  Ours  |        |
> |----------|:-------:|:------:|:------:|:-------:|:------:|:------:|
> |          | Uniform | beta=1 | change | Uniform | beta=1 | change |
> |   IoU↑   |  0.934  |  0.902 | -0.032 |  0.953  |  0.954 | +0.001 |
> | Chamfer↓ |  0.041  |  0.050 | +0.009 |  0.040  |  0.040 | +0.000 |
> |    F1↑   |  0.967  |  0.937 | -0.030 |  0.966  |  0.964 | -0.002 |
>
>
> > In Figure 4, what’s the output of the model? How is a point cloud or another shape representation decoded from it?
>
> This figure is to illustrate the autoregressive generation of discrete tokens.
>
> * Training
>
> 	* The inputs are conditional information C and discrete tokens. The tokens are obtained from the vector quantization (Figure 3 Lower Right and Eq (5)).
> 	* The outputs are the probability of discrete tokens (Eq (8))
>
>
> * Inference
>
>
> 	* In each sampling step, we feed the conditional information C and sampled token set to the network (Eq (8)).
>
> 	* In each sampling step, a discrete token is sampled from the output probability distribution.
>
> When we have all tokens, we decode the discrete tokens to latents and then final implicit surfaces (Figure 3)
>
>
> The pipeline is commonly used in VectorQuantization+AutoRregressive models (e.g., VQVAE [1], VQGAN [2], AutoSDF [3] and ShapeFormer [4]).
>
> > Figure 5 is inconsistent with Figure 4. Show a block diagram of the proposed architecture, as you do in Figure 4.
>
> The training of the two methods is very different. Figure 4 shows an architecture that generates tokens one by one in an autoregressive fashion. Figure 5 shows an architecture that fills in / replaces masked tokens in a given grid of tokens. Some of the replaced tokens are kept and others (low probability ones) discarded. The discarded tokens are again masked and the process proceeds iteratively.
>
> > In Section 4, how are x_{i,k} obtained for different i-s? What is the difference between x_{i,k} and x_{j,k} for different i, j? Please explain.
>
> X_i is a 3D coordinate, x_{i,1}, x_{i,2} and x_{i,3} are 3 components of the coordinate. This is explained in Sec 4.1 L157. The coordinates x_i are obtained from FPS, which is explained in L121 and Eq (1). Both x_i and x_j are the points belonging to the subsampled point set in Eq (1).

---

> > ### Author Response · Authors · 2022-08-02
> > **Response to Reviewer jKmw Part 2**
> >
> >
> > > In Section 5, further comparison with IF-NET is needed: the proposed method slightly improves over IF-NET. How does it compare to IF-NET (and other methods) in other respects - resource consumption, train and inference time, etc.?
> >
> >
> >
> > > In A.4., how do the methods compare in terms of memory and runtime?
> >
> > We measure the memory consumption and runtimes of ConvOccNet and IF-Net. The metrics are compared with two variants of our method (Nadaraya–Watson estimator and knn interpolation). The occupancies are evaluated on a 128x128x128 grid. Then, we perform MarchingCubes on the grid to get triangle meshes. For both IF-Net and Ours(NW), we are unable to fit the 128x128x128=2097152 points to an 80GB A100 GPU. So we feed 50k points in a forward pass. The results can be found in the row Multiple Pass. The quality would remain the same for Multi Pass and Single Pass. The runtime is comparable, but our memory consumption is higher.
> > |                |             | ConvOccNet | IF-Net | Ours(NW) | Ours(KNN) |
> > |----------------|-------------|:----------:|:------:|:--------:|:---------:|
> > | Multiple  Pass |  Time(sec)  |   0.3119   | 0.5743 |   2.03   |   0.3367  |
> > |                | Memory (MB) |     500    |  1600  |   25343  |    509    |
> > |  Single  Pass  |  Time(sec)  |    0.104   |    -   |     -    |   0.302   |
> > |                | Memory (MB) |    1359    |    -   |     -    |   17689   |
> >
> > > Explain the purpose of Eq.(8) and Eq.(9).
> >
> > Eq (8) is the likelihood of generation we are going to maximize. Eq (9) is the detailed expansion of each term on the right hand of Eq (8).
> >
> >
> >
> > > In A.6., ll.62-63 - “Bad” in what sense? Non-realistic generations by the proposed method should be highlighted as well.
> >
> > Examples of undesirable results are: 1) Missing thin structures. 2) Overly blurred / smoothing of sharp features or smaller details. 3) Creation of spurious and unnatural details. We highlighted more samples as requested in the current revision.
> >
> > > In l.130, explicitly state which positional encodings are used, for completeness of the presentation.
> >
> > We followed the suggestion and now formulated the positional encodings in the appendix A.11.
> >
> > > Introduction section needs to be revised to more clearly describe the main aspects of the proposed approach.
> >
> > We will make it more clear in the final revision.
> >
> > > Missing references
> >
> > We have added the mentioned references in the current revision.
> >
> > > Nit: In Table 1, consider writing all numbers in the same manner such that it will be clear which parameters are identical for different methods.
> >
> > We discussed this suggestion, but would prefer to keep the numbers as is. Otherwise, it is hard to understand the grid resolutions, if the numbers are multiplied.
> >
> > ### References
> >
> > [1] Van Den Oord, Aaron, and Oriol Vinyals. "Neural discrete representation learning." Advances in neural information processing systems 30 (2017).
> >
> >
> >
> > [2] Esser, Patrick, Robin Rombach, and Bjorn Ommer. "Taming transformers for high-resolution image synthesis." In Proceedings of the IEEE/CVF conference on computer vision and pattern recognition, pp. 12873-12883. 2021.
> >
> >
> >
> > [3] Mittal, Paritosh, Yen-Chi Cheng, Maneesh Singh, and Shubham Tulsiani. "Autosdf: Shape priors for 3d completion, reconstruction and generation." In Proceedings of the IEEE/CVF Conference on Computer Vision and Pattern Recognition, pp. 306-315. 2022.
> >
> >
> >
> > [4] Yan, Xingguang, Liqiang Lin, Niloy J. Mitra, Dani Lischinski, Daniel Cohen-Or, and Hui Huang. "Shapeformer: Transformer-based shape completion via sparse representation." In Proceedings of the IEEE/CVF Conference on Computer Vision and Pattern Recognition, pp. 6239-6249. 2022.

---

### Official Review · Reviewer_uFdG · 2022-07-17

**Rating:** 4
**Confidence:** 4
**Soundness:** 3 good
**Presentation:** 3 good
**Contribution:** 2 fair

**Summary:**

This paper proposes a geometry representation based on irregular grids for generative shape modeling.

**Questions:**

My suggestions for improvement are:
- Having examples with non-shape-net data to illustrate robustness and utility in practice.
- Having comparisons with surface reconstruction techniques other than only neural grids. The point clouds look clean and seem to sample the shapes well.
- If no good results with real data for generative tasks, I would drop that claim and focus on the new representation's ability for reconstruction, efficiency, spatial adaptivity, etc.

**Limitations:**

Please see above.

**Strengths And Weaknesses:**

Strengths:
- It is an interesting idea, especially when combined with transformers and vector quantization.
- Some good results on reconstructed and generated shapes, although I am not convinced with the comparisons.

Weaknesses:
- The advantage over point cloud-based generative modeling is not well established, both conceptually and experimentally.
- It is a bit of a stretch to call this a grid-based algorithm and only consider grid-based alternatives.
- The reconstructions are only compared against other networks, there are no comparisons to surface reconstruction methods e.g. Poisson or edge-preserving MLS.
- The proposed generative tasks are only on shape-net, no real images or point clouds.
- Memory and runtimes are not clear.

---

> ### Author Response · Authors · 2022-08-02
> **Response to Reviewer uFdG Part 1**
>
> >The advantage over point cloud-based generative modeling is not well established, both conceptually and experimentally.
>
>
> We agree that different 3d representations (e.g., point clouds, voxels, implicit surfaces) have their own advantages and disadvantages. Our work is an implicit surface generation method (via neural fields, often called neural implicit representations or coordinate-based networks). When we have an implicit surface, the corresponding point cloud representation can be obtained by sampling on the surface. However, obtaining a surface-based representation for a given point cloud is much more difficult and the process may result in errors. In general, a surface-based representation (such as ours) contains more information about topology and local neighborhoods and it is therefore more appropriate for downstream tasks, such as rendering, analysis, and editing. Therefore, we believe it would be unusual to compare surface-based representations to point-cloud based representations in actual experiments. While the specific advantage over point-based generative modeling may not be established in our paper, the general advantage of surfaces over point-clouds is well established in general. We therefore do not think that pure point-based modeling can be seen as a competitor.
>
>
>
> A distinct advantage of modern data structures for generative modeling is that they interpolate high-dimensional features and then post-process the high-dimensional feature using an MLP. If the suggestion is to adapt point clouds to be used as backbone of a neural field, there are multiple ingredients that are unspecified. How to interpolate features from a point cloud, how to incorporate high-dimensional features, and how to integrate everything so it becomes trainable end to end. This is a research project on its own and requires multiple design decisions to be made. In particular, it seems difficult to integrate Poisson Surface Reconstruction in such a framework. While recent research from NeurIPS 2021 shows how to make Poisson Surface Reconstruction differentiable, this still requires quite a bit of work to build a complete system and it is not evident how it could be competitive. It seems much more intuitive to build on Neural Fields. We will clarify this in the final revision.
>
> > It is a bit of a stretch to call this a grid-based algorithm and only consider grid-based alternatives.
>
>
>
> The focus of our work is to study shape representations in the context of generative modeling based on neural fields. Again, it is important that a data structure is compatible with generative modeling, e.g. can be trained with a VQ-VAE and be used by an auto-regressive transformer. Further, the data structure / framework needs to have the ability to interpolate high-dimensional latents post-processed with an MLP and the framework needs to be trainable end-to-end including the interpolation. In this context, there are no other competing data structures and we compare to the state of the art that uses regular grids (ConvOccNet [1]) and a multi-scale grid (IF-Net [2]). Also, 2D grids are a popular representation for generative image modeling and many state of the art methods build on 2D grids, e.g. Parti, Dalle2, VQGAN. This also gives additional support to our claim that it is reasonable to assume that 3D grids would be the current state of the art.
>
>
>
> > The reconstructions are only compared against other networks, there are no comparisons to surface reconstruction methods e.g. Poisson or edge-preserving MLS.
>
> >Suggestion: Having comparisons with surface reconstruction techniques other than only neural grids. The point clouds look clean and seem to sample the shapes well.**
>
> Poisson Surface Reconstruction is only a small part of a potential competing framework. If the suggestion is to use Poisson Surface Reconstruction as an independent post-process combined with a generative method to generate point clouds (not sure which one), there are many open design decisions and this would require additional research as discussed above. If the suggestion is simply to compare only the reconstruction part, this is a good idea to make the paper more complete and we are happy to provide this additional information below.
>
> We compare our method with Poisson Surface Reconstruction (PSR) [3]. The results are shown in the following Table. Best results are highlighted in **bold** and second best results are shown in _italic_. Note that PSR requires per-point normals as input. We also updated the results in the current revision (Appendix A.12). In summary, Poisson Surface Reconstruction performs well, but not as well as our method or IF-Net.
>
> |           |  PSR  | OccNet | ConvOccNet | IF-Net | Ours(3DILG) |
> |-----------|:-----:|:------:|:----------:|:------:|:-----------:|
> | Chamfer ↓ | 0.043 |  0.072 |    0.052   |  _0.041_ |    **0.040**    |
> | F-Score ↑ | 0.922 |  0.858 |    0.933   |  **0.967** |    _0.966_    |

---

> > ### Author Response · Authors · 2022-08-02
> > **Response to Reviewer uFdG Part 2**
> >
> > > The proposed generative tasks are only on shape-net, no real images or point clouds.
> >
> > > Suggestion: Having examples with non-shape-net data to illustrate robustness and utility in practice.
> >
> > > If no good results with real data for generative tasks, I would drop that claim and focus on the new representation's ability for reconstruction, efficiency, spatial adaptivity, etc.
> >
> > We clarify that, we already have an additional experiment on scene level reconstruction in the original submission which is shown in the appendix A.4. However, it is true that more additional datasets on real data could improve the paper. We therefore conducted more experiments as follows:
> >
> > 1.  We show real world images as conditional input to our generative model trained on ShapeNet. The images are from the dataset ABO [4]. The results can be found in the Appendix A.7 (Fig. 17).
> >
> > 2.  We show real world scans reconstruction results with our reconstruction network trained on ShapeNet. These scans are taken from the dataset D-FAUST [5]. We evaluate both IF-Net and our method. The results can be found in the following table. Also see Appendix A.8 for a visualization of reconstructed meshes.
> >
> > |           | IF-Net | Ours   |
> > |-----------|--------|--------|
> > | Chamfer ↓ | 0.0238 | **0.0210** |
> > | F-Score ↑ | 0.9940 | **0.9953** |
> >
> > 3.  We can run our methods (category-conditioned generation) on the dataset ABO [4] from scratch. The results can be found in the Appendix A.9 (Fig. 19). Note that due to the short time limit of the rebuttal, we only choose a subset of the dataset ABO to train our network. In this experiment, the train/val/test set are composed of 1184 chair models in total. The size is much smaller than ShapeNet (52472).
> >
> >
> > > Memory consumption and running times are not clear.
> >
> > We measure the memory consumption and runtimes of ConvOccNet and IF-Net. The metrics are compared with two variants of our method (Nadaraya–Watson estimator and knn interpolation). The occupancies are evaluated on a 128x128x128 grid. Then, we perform MarchingCubes on the grid to get triangle meshes. For both IF-Net and Ours(NW), we are unable to fit the 128x128x128=2097152 points to an 80GB A100 GPU. So we feed 50k points in a forward pass. The results can be found in the row Multiple Pass. The quality would remain the same for Multi Pass and Single Pass. The runtime is comparable, but our memory consumption is higher.
> >
> > |                |             | ConvOccNet | IF-Net | Ours(NW) | Ours(KNN) |
> > |----------------|-------------|:----------:|:------:|:--------:|:---------:|
> > | Multiple  Pass |  Time(sec)  |   0.3119   | 0.5743 |   2.03   |   0.3367  |
> > |                | Memory (MB) |     500    |  1600  |   25343  |    509    |
> > |  Single  Pass  |  Time(sec)  |    0.104   |    -   |     -    |   0.302   |
> > |                | Memory (MB) |    1359    |    -   |     -    |   17689   |
> > ### References
> > [1] Peng, Songyou, Michael Niemeyer, Lars Mescheder, Marc Pollefeys, and Andreas Geiger. "Convolutional occupancy networks." In European Conference on Computer Vision, pp. 523-540. Springer, Cham, 2020.
> >
> > [2] Chibane, Julian, Thiemo Alldieck, and Gerard Pons-Moll. "Implicit functions in feature space for 3d shape reconstruction and completion." In Proceedings of the IEEE/CVF Conference on Computer Vision and Pattern Recognition, pp. 6970-6981. 2020.
> >
> > [3] Kazhdan, Michael, Matthew Bolitho, and Hugues Hoppe. "Poisson surface reconstruction." In Proceedings of the fourth Eurographics symposium on Geometry processing, vol. 7. 2006.
> >
> > [4] Collins, Jasmine, Shubham Goel, Kenan Deng, Achleshwar Luthra, Leon Xu, Erhan Gundogdu, Xi Zhang et al. "Abo: Dataset and benchmarks for real-world 3d object understanding." In Proceedings of the IEEE/CVF Conference on Computer Vision and Pattern Recognition, pp. 21126-21136. 2022.
> >
> > [5] Bogo, Federica, Javier Romero, Gerard Pons-Moll, and Michael J. Black. "Dynamic FAUST: Registering human bodies in motion." In Proceedings of the IEEE conference on computer vision and pattern recognition, pp. 6233-6242. 2017.

---

### Author Response · Authors · 2022-08-02
**To All Reviewers**

We thank the reviewers for their comments. We are happy to see that reviewers found our papers to bring an “**interesting**” idea to 3D generation (Reviewer uFdG and tLeQ), that our method is “**novel**” (Reviewer jKmw) and explores “**sparse nature**” of 3D shapes (Reviewer tLeQ). We are also glad our work was found to show results which are “**impressive, accurate, detailed**” (Reviewer jKmw), “**comprehensive**” (Reviewer jKmw), “**promising**” (Reviewer tLeQ), and “**qualitatively good**” (Reviewer uMXb). Finally, we are delighted to see that our technical presentation is “**excellent**” (Reviewer uMXb), and “**clear**” (Reviewer tLeQ and Reviewer jKmw). We would like to post a summary of all new experiments for this rebuttal here and answer to the individual reviewers separately.



### Summary of new experiments

1.  Comparison with traditional surface reconstruction, Poisson Surface Reconstruction. (Reviewer uFdG)

2.  Generalization results on real point clouds, D-FAUST human models (Reviewer uFdG)

3.  Generalization results on real world images conditioned generation, ABO images (Reviewer uFdG)

4.  Generative model trained on ABO models (Reviewer uFdG)

5.  Memory and runtime comparison (Reviewer uFdG and jKmw)

6.  Reconstruction performance on non-uniformly sampled point clouds (Reviewer jKmw)

7.  Alternative choice of interpolation method (Reviewer uMXb)

---

### Author Response · Authors · 2022-08-08
**To All Reviewers**

Dear Reviewers,

Thanks again for the review. Since the author/reviewer discussion phase is ending tomorrow, we would like to ask if our comments helped clarify your concerns or if there are additional questions we can help with.

---

### Meta-Review · Area_Chair_9HRv · 2022-08-24

**Recommendation:** Accept
**Confidence:** Certain

**Metareview:**

All reviewers agree to accept this work, which presents a creative new shape representation for 3D generative modeling.  The negative aspects raised by the reviewers are fairly minor, and most were addressed during the rebuttal phase (please be sure to incorporate all comments/additional results into the final camera-ready version).  During the post-rebuttal discussion, reviewers suggested nominating for a spotlight given the new results on realistic data.

**Award:**

Yes

---

### Decision · Program_Chairs · 2022-09-14

Accept